# Negative Acts in the Courtroom: Characteristics, Distribution, and Frequency among a National Cohort of Danish Prosecutors

**DOI:** 10.3390/bs14040332

**Published:** 2024-04-16

**Authors:** Amanda Ryssel Hovman, Jesper Pihl-Thingvad, Ask Elklit, Kirsten Kaya Roessler, Maria Louison Vang

**Affiliations:** 1The Danish Center of Psychotraumatology, Department of Psychology, University of Southern Denmark, Campusvej 55, 5230 Odense, Denmark; 2Department for Occupational and Environmental Medicine, Odense University Hospital, 5000 Odense, Denmark; 3Department of Psychology, University of Southern Denmark, Campusvej 55, 5230 Odense, Denmark; 4Department of Clinical Research, University of Southern Denmark, 5000 Odense, Denmark

**Keywords:** negative acts, counterproductive workplace behavior, incivility, aggression, workplace bullying

## Abstract

Danish prosecutors report exposure to negative acts from professional counterparts in courtrooms, which is associated with an increased risk of burnout. However, knowledge of the characteristics of these acts is limited. Based on existing theoretical frameworks, this study aims to characterize these negative acts. A nation-wide survey of Danish prosecutors (response rate: 81%) yielded 687 descriptions of experiences with negative acts from professional counterparts from a career perspective. These were analyzed using theory-directed content analysis based on the Stress-as-Offense-to-Self (SOS) theory by Semmer and colleagues and Cortina and colleagues’ characterization of incivility in American courtrooms. We identified a total of 15 types of behavior within the three main themes: illegitimate tasks (n = 22), illegitimate stressors (n = 68), and illegitimate behavior (n = 612). Tentative differences in the distribution of experienced negative acts from a career perspective were found for gender and seniority. Women reported negative acts more frequently than men, and assistant prosecutors reported verbal abuse more frequently than senior prosecutors, who, conversely, more often reported a perceived lack of court management. More prospective research is needed on negative acts experienced by prosecutors to assess the scope of these in Danish courtrooms and how they impact the risk of burnout.

## 1. Introduction

Prosecutors play a vital role in the Danish legal system, as their core task within the Danish Prosecution Service is to prosecute criminal offenders while avoiding the conviction of innocent people [1]. The prosecutors act on behalf of the Prosecution Service as an institution rather than as individuals. They are the only professionals in Denmark allowed to appear before court and press charges under the Danish Administration of Justice Acts (1919) [2]. According to the 2022 EU Justice Scoreboard [3], Denmark took first place out of 26 European countries regarding the number of incoming court cases per 100 inhabitants in the years 2012–2020. Conversely, Denmark places second to last among 27 European countries in general governmental expenditures for courts of law, measured by the percentage of GDP and the number of judges per 100,000 inhabitants [3]. Therefore, the Danish legal system is at risk of promoting a working environment characterized by a lack of resources for judges and prosecutors, who are governmentally funded. The average waiting time before a criminal case has been heard by a lay judges’ court has increased from 4.4 months in 2018 to 8.3 months in 2022 [4]. Work environmental factors such as high work pressure can increase the risk of adverse occupational mental health outcomes, including stress-related mental disorders and the exhaustive component of burnout [5].

### 1.1. Occupational Mental Health among Prosecutors

International research on attorneys indicates that working with trauma-exposed clients increases the risk of developing job burnout and symptoms of post-traumatic stress disorder (PTSD) and secondary traumatization [6,7,8]. Additionally, working with traumatized clients may elevate lawyers’ risk of developing depression, anxiety, and vicarious trauma, as found by Rønning and colleagues [9]. In a study on occupational stress among public prosecutors in Pakistan, Bilal and Batool [10] report work strain and lowered job satisfaction among public prosecutors experiencing work stressors such as a lack of facilities, performance pressure, and negative acts from professional counterparts. Thus, prosecutors are exposed to both quantitative and emotional demands. According to Maslach and colleagues [11], quantitative demands such as a high caseload and not having enough time for one’s work tasks are strongly correlated to burnout, and correlations have also been found between burnout and qualitative demands such as role conflicts. This suggests that prosecutors are also at risk of experiencing burnout [5,12,13,14]. Vang and colleagues [14] were the first to target the occupational mental health of prosecutors in Denmark. They studied the extent and correlates of occupational mental health conditions among prosecutors and did a nation-wide online survey with a response rate of 39%. Here, they found almost half of the responding prosecutors (47.2%) to be at risk of experiencing burnout. The main correlates of burnout were quantitative demands, negative acts from professional counterparts, and emotionally demanding casework, listed in order of importance [14]. On average, Danish prosecutors reported ‘sometimes’ or ‘often’ being under time pressure or having a higher workload than manageable within ordinary working hours, reflecting elevated quantitative demands. Regarding the negative emotional impact of dealing with cases at work, 42.7% of the Danish prosecutors reported having worked with cases during the last year that left a strong emotional impression on them [14]. Thus, existing research suggests that the occupational mental health of lawyers can be negatively affected by their working environment and work content involving emotionally demanding work tasks [5,10,11,13,14].

### 1.2. Negative Acts among Legal Professionals

Negative acts in the workplace have been described in terms such as bullying, harassment, and incivility [15,16,17,18]. Being exposed to negative acts in the workplace increases the risk of developing anxiety, depression, burnout, and PTSD [16,17,19,20], thus affecting the mental health of the exposed employee. Negative acts have been found to exacerbate the effect of job demands on depression and uncertified absenteeism, as Devonish [21] found employees at higher risk of physical exhaustion, depression, and voluntary absenteeism when they were experiencing high levels of both job demands and workplace bullying at the same time [16,21]. As such, negative acts in the workplace are considered a symptom of high-strain employment and can increase the risk of stress reactions, such as burnout beyond the risk incurred by quantitative demands [15,16,17,18,20]. Vang and colleagues [14] found negative acts from professional counterparts to be the second-strongest predictor of burnout among Danish prosecutors, with 27% of surveyed Danish prosecutors having been exposed to aggressive or negative acts from a professional counterpart [14]. As such, negative acts from professional counterparts are not a trivial problem for public prosecutors, considering both the frequency and potential consequences of exposure to these acts. This is in line with international research where Omari and Paull [22] surveyed 327 members of an association of legal professionals in Australia (response rate of 12%), examining organizational culture, climate, and experiences with workplace bullying. They found reports of negative acts ranging from exclusion, inappropriate comments, and verbal behaviors, such as being talked down to and belittled, to aggressive approaches and throwing objects. Omari and Paull [22] point to the legal profession as a high-risk environment for the occurrence of bullying, as the profession is characterized by traditions of hierarchy, power, and status and a demanding nature of work practices.

### 1.3. Existing Knowledge of Negative Acts among Legal Professionals

Existing knowledge of negative acts among legal professionals mainly stem from cross-sectional studies that do not examine the characteristics of negative acts occurring in court rooms [14,22]. As such, existing knowledge on negative acts in the workplace for this profession is limited [22] and so is our understanding and practical possibilities for intervening in the relationship between courtroom aggression and its impact on the occupational mental health of prosecutors and other legal professionals. Hence, with the current study, we aim to characterize the negative acts in the workplace reported by Danish prosecutors to further broaden our knowledge of these acts within courtrooms. This will help inform future prevention strategies targeting negative acts in court rooms.

## 2. Framework for Characterizing the Negative Acts Reported by Danish Prosecutors

### 2.1. Stress-as-Offense-to-Self

The Stress as Offense to Self (SOS) framework has previously proved useful in understanding the detrimental impact of negative acts in the workplace, considering how negative acts affect the professional identity of employees [23,24]. Employees assume a professional identity at work, which is a valued part of their global self-view and includes a set of expectations, roles, and work tasks. If the professional identity is challenged or violated, both the professional identity and the self-view may be threatened, resulting in employee stress reactions [24]. One such threat might be negative acts from professional counterparts [24]. Three types of stressors are presented within the SOS framework: *illegitimate tasks*, *illegitimate stressors*, and *illegitimate behavior* [24].

*Illegitimate tasks* consist of work tasks that are not considered part of one’s professional identity and assigned tasks and are considered either unnecessary or unreasonable. Unnecessary tasks are tasks that could be avoided, for example, by changing the procedures for work. Therefore, one might see performing such tasks as an unnecessary nuisance [24]. Unreasonable tasks are tasks not considered to be part of one’s job function and should be performed by someone with another job type [24].

*Illegitimate stressors* do not have to be intentionally directed against the recipient but are acts perceived to involve a lack of consideration for the interests of others [23]. Stressors are illegitimate if they are caused or aggravated by behavior perceived as inconsiderate [23] or if they are not considered typical for one’s profession and therefore might be considered threatening to the employee’s professional identity [25].

*Illegitimate behavior* is a direct way of showing disrespect for others and consists of deliberate and disrespectful behavior such as ridiculing others, making them lose face, or subjecting them to unfair or unreasonable feedback [23]. Illegitimate behavior has previously been investigated in the context of social stressors under the concept of workplace bullying but has received limited attention. Rather than seeing workplace bullying and negative acts as isolated stressors, Semmer and colleagues [23] argue that illegitimate behaviors have a strong connection to occupational stress in general [23,25].

In attending court, the prosecutors participate in a working environment that differs from that of their organization and regular workplace. Here, judges are tasked with courtroom management [1,26,27]. Despite a lack of clear definitions of the tasks involved in courtroom management, several users of the Danish courtroom, including defense lawyers and prosecutors, have stated how they expect the judges to exercise an explicit, steadfast, and respectful courtroom management style towards all parties present [26]. Therefore, we would hypothesize that judges assigning illegitimate tasks to prosecutors or accepting negative acts from defense lawyers without intervening might be perceived by prosecutors as engaging in differential treatment between defense lawyers and themselves. Thus, illegitimate tasks and stressors could be perceived by the prosecutors as intentional, recurring, and directed against them by both defense lawyers and judges, giving these types of behaviors the status of negative acts.

### 2.2. Illegitimate Behavior and Incivility

In a recent modification of the SOS theory, Semmer and colleagues [25] further emphasized how illegitimate behavior is also reflected in existing research on incivility, a concept introduced by Andersson and Pearson [28]. Incivility is defined as low-intensity, deviant behaviour, violating the workplace norms for mutual respect with ambiguous intent to harm the target [28]. This framework on incivility has been used in previous international research on legal professionals, for example in a study of 4608 American attorneys (53% response rate) by Cortina and colleagues [29]. They examined the prevalence, nature, and interplay of negative acts in the form of incivility experienced by attorneys in courtrooms and found that incivility varies in severity from mild discourtesy to extreme hostility. Overall, Cortina and colleagues [29] found that 62% of the participating attorneys had experienced some form of interpersonal mistreatment during litigation in the previous 5 years. The authors identified 11 types of behaviours using an iterative coding process on 483 prosecutors’ replies to open-ended questions about the experience with interpersonal mistreatment in the past 5 years of their career that made the greatest impression upon them: *disrespect/dishonesty*, *ignoring/exclusion*, *professional discrediting*, *silencing*, *gender disparagement*, *threats/intimidation*, *unprofessional address*, *appearance comment*, *mistaken identity*, *sexualised behaviour*, and *touching*. Cortina and colleagues [29] found that general incivility, including disrespect/dishonesty, ignoring/exclusion, professional discrediting, silencing, and threats/intimidation, was the most frequent form of incivility in federal courts. This was followed by gender-related incivility, including unprofessional forms of address, comments on appearance, and mistaken identity. Taken altogether, existing research [23,28,29] suggests that SOS and incivility might be useful frameworks to further describe and understand the negative acts from professional counterparts in Danish courts of law.

## 3. Aim of the Current Study

The current study has two aims. The first is to describe and characterize the types of negative acts experienced by Danish prosecutors using theory-directed content analysis informed by the SOS theory and the concept of incivility [23,29]. Secondly, we aim to explore the source and distribution of these acts among Danish prosecutors across demographic characteristics.

## 4. Method

### 4.1. Procedure

Based on previous research indicating a relationship between negative acts from professional counterparts and burnout [16], a survey was developed to study the types of negative acts encountered by Danish prosecutors. These were mapped via open-ended questions for prosecutors to describe up to three instances of uncomfortable or negative acts encountered by their professional counterparts at any point during their career. Open-ended answers were prompted by asking participants to answer a dichotomous (yes/no) question: Have you, at some point during your career, had any uncomfortable experiences with your professional counterparts? Here, we refer to incidents that you have found to be uncomfortable in connection with conducting cases in court (e.g., verbal abuse; a condescending attitude, offensive, derogatory, or devaluating personalized comments; disrespectful remarks or acts; or the like). Participants answering affirmatively were further prompted to describe up to three experiences by the following question: “We would like you to shortly describe up to three different incidents with verbal abuse or uncomfortable behavior from your professional counterparts (judges and defense lawyers) that you have experienced at some point during your career”. There were no character limits to their description. Data was collected via an online anonymous survey in the period from 10 August 2021 to 12 October 2021. All prosecutors employed by the Danish Prosecution Service during that time were invited to participate. Invitations were sent to the prosecutors’ work email via the office of the Director of Public Prosecutions for Education and Development and via QR codes distributed to participants at the Annual National Conference for Prosecutors on 23 September 2021, where the last author gave a presentation on the project. The survey was hosted by the University of Southern Denmark, which was the data controller, and distributed via SurveyXact. Data was handled in accordance with General Data Protection Regulation (GDPR), and approval for the surveys was granted from the data protection office at the University of Southern Denmark under application number 11,395. Informed consent was obtained from all participants involved in the study.

### 4.2. Participants

A total of N = 572 prosecutors participated in the survey, corresponding to a response rate of at least 81%. There is uncertainty surrounding the computation of the response rate due to the period of data collection covering 3 months during which many prosecutors started employment, and it is unclear whether they received the invitation or reminders during their start at work. We have used the largest estimate of population N from the personnel office of the Director of Public Prosecutions (N = 700) to compute the participation rate, which is therefore a conservative estimate. Most participants were women (n = 369, 66.6%) within both the professional group of assistant prosecutors (n = 192, 70.6%) and prosecutors of higher seniority (n = 428, 70.4%), and the average age was 37.5 years (SD = 10.55). A total of n = 244 prosecutors (44.2%) were employed in the Eastern Danish Prosecution Service, n = 194 (35.1%) were employed in the Western Danish Prosecution Service, and n = 103 (18%) were employed in the Central Danish Prosecution Service. The mean seniority was 9.3 years (SD = 9.49), and approximately 20% (n = 111) were responsible for mentoring more inexperienced colleagues. The most frequently represented job title was assistant prosecutor (n = 187, 33.9%), followed by prosecutor (n = 159, 28.9%), senior prosecutor (n = 113, 19.8%), chief prosecutor (n = 10, 1.7%), and managing attorneys (n = 25, 4.4%).

### 4.3. Data Analysis

*Directed content analysis (aim 1):* The characterization of negative acts experienced by Danish prosecutors was based on a directed content analysis [30], conducted by the first and last author. The analysis included open-ended responses from the preliminary survey. We began our analysis with predetermined codes based on the three types of stressors introduced by the SOS theory. Any data that could not be meaningfully coded within this framework was identified and analyzed later to determine if the data represented a new category or could be considered a subcategory within an existing code. The codes identified by Cortina and colleagues [29] were used as a guide for developing subcodes under the *illegitimate behavior* theme. All relevant coding categories were applied to each individual description, ensuring that the data was thoroughly coded. Thus, several codes may be applied to the same description if it contains multiple types of negative acts. Therefore, the reported percentages of each coding category will not add up to 100%.

The directed content analysis consisted of five steps:The first and last author independently read the case material in its full length and continuously coded the data informed by the SOS theory.The first and last author discussed the initial coding to characterize the negative acts reported by the prosecutors. This included a discussion of any uncoded data that was either grouped with an existing code or given its own additional category. At this stage, codes from Cortina and colleagues [29] were added to the coding tree, and a total of ten subcodes of illegitimate behavior were added to refine this category.The first author recoded the data, focusing on the codes from Cortina and colleagues [29], distributing the data within the *illegitimate behavior* to these subcodes.The last author perused a random sample of the coded case material in accordance with the coding scheme. Upon disagreement, challenges related to the clarity or applicability of the categories or to the match between categories and data were discussed until agreement. This process was repeated three times.The first and last author met and finalized the recoding of the data to determine the coding of the individual descriptions. Disagreements were discussed until resolved.

*Distribution of negative acts (aim 2):* The distribution of negative acts across judges and defense lawyers was analyzed using descriptive statistics. Differences in the distribution of negative acts across demographic characteristics of prosecutors (men/women, junior/senior employees, and geographical region) were analyzed using Pearson Chi^2^ analysis for expected cell counts larger than 5. For expected counts less than 5, the Fisher’s exact test was applied. Adjusted standardized residuals are reported for all Chi^2^ analyses to identify directions of differences in the distribution of negative acts. For adjusted standardized residuals exceeding −2.00 and 2.00, we assumed there to be a difference in the distribution within the specific category, with negative values showing fewer observed reports of a negative act than expected and vice versa for positive values [31]. Due to small sample sizes (<5%), participants (n = 7, 1.3%) who did not want to disclose their gender or chose ‘other gender identity’ were excluded from further analysis of the distribution of reports on negative acts. This also applies to participants (n = 11, 2%) employed in Greenland and the Faroe Islands. When performing the statistical analysis on seniority, we combined prosecutors, senior prosecutors, chief prosecutors, and managing prosecutors in one overall category named ‘prosecutor or higher’, as some of these subgroups had sample sizes <5%. All statistical analyses were performed using IBM SPSS Statistics (Version 28), with Fisher’s Exact Test for tables exceeding the 2 × 2 format performed using R Statistical Software (v4.3.2, [32]) and the gmodels R package [33]. Chi^2^ analysis for the distribution of negative acts across geographical regions can be found in the Appendix A.

## 5. Results

Of the 572 participating prosecutors in the present study, 366 reported having had an unpleasant experience with a professional counterpart at some point during their career, corresponding to 64.0% of the sample. The analysis conducted in the present study was based on 687 descriptions of experiences with negative acts reported across n = 350 (61.19%) Danish prosecutors. A total of 49 descriptions were excluded from further analysis due to the content of the descriptions or the reported source. This included descriptions of negative acts from unknown sources, sources outside the courtroom, or from sources other than professional counterparts, as these would not add to our understanding of negative acts from professional counterparts. General descriptions of the overall tone of voice or other environmental factors were also excluded, as they could not support an exploration of the specific types of negative acts reported by prosecutors.

The directed content analysis resulted in a total of 15 codes for negative acts, based on both the SOS theory and the construct of incivility [23,24,29]. Three of these were the main coding categories from the SOS theory: *illegitimate tasks*, *illegitimate stressors*, and *illegitimate behavior*, within which the remaining 12 codes were distributed. Cortina and colleagues’ [29] codes of *ignoring/exclusion* and *silencing* were combined, and *age* was added to the code of *gender disparagement*. Hence, gender-related incivility was renamed person-focused incivility, thereby reflecting personalized disparagement. Table 1 presents the identified main and subcodes, with quotes exemplifying the content of each subcode.

The coding category *illegitimate tasks* was applied to 3.2% (n = 22) of the descriptions. Illegitimate tasks included descriptions of experiences with being asked, expected to, or made to perform task assignments that are outside of one’s job description and therefore are perceived as either unnecessary or unreasonable. For example, these described disinfecting the desks and tables within the courtroom, moving chairs around, and getting water for the accused. Illegitimate tasks included no sub-codes. *Illegitimate stressors* included accounts of stressors that are not considered typical for the prosecutors’ profession or are caused by the inconsiderate behavior of others and were applied to 9.9% (n = 68) of the descriptions. Within this coding category, two subcodes were identified: *lack of court management* (n = 30) and *conflicts arising or exaggerated by illegitimate stressors* (n = 38). Lack of court management included descriptions of judges failing to intervene in situations with inappropriate behavior. Conflicts arising or exaggerated by illegitimate stressors are described, for example, as being held accountable for issues arising outside one’s area of responsibility or being yelled at by judges due to the faults of others. Of the three main coding categories, *illegitimate behavior* was the largest and applied to 89.1% (n = 612) of the descriptions of negative acts. This category described experiences with behavior that explicitly shows disrespect and included the following three subcodes: *verbal abuse* (n = 124), *general incivility* (n = 483), and *person-focused incivility* (n = 48). The types of behavior ranged in severity from high-arousal situations like yelling at the prosecutor in front of the court to more subtle, discourteous behavior such as eye-rolling. The category ‘verbal abuse’ described experiences with high arousal, verbal abuse, raised voices, and yelling and included no additional subcodes. General incivility was the largest subgroup and included the four following additional subcodes: *disrespect/dishonesty* (n = 308), *professional discrediting* (n = 142), *threats/intimidation* (n = 47), and *ignoring/exclusion/silencing* (n = 21). Of these, disrespect/dishonesty was the largest subcategory and described experiences with, e.g., condescending tones of voice, interruptions during procedures, and unreasonable feedback given in public. Professional discrediting included having one’s professional capabilities questioned or being accused of withholding evidence. Threats/intimations included being threatened with complaints and reports to the Independent Police Complaints Authority or being intimidated by defense lawyers during procedures. Ignoring/exclusion/silencing was a combination of the codes ignoring, exclusion, and silencing from Cortina and colleagues [29] and included descriptions of being excluded from conversations, being silenced by professional counterparts through interruptions, or being told off. Person-focused incivility included three additional subcodes, as well as implementations or modifications of the codes introduced by Cortina and colleagues [29]: *gender/age disparagement* (n = 36), *appearance comments* (n = 8), and *unprofessional address* (n = 7). Age was added to Cortina and colleagues’ [29] code of gender disparagement, resulting in the code of gender/age disparagement holding descriptions of comments on or associated with the age or gender of the prosecutor. Appearance comments included descriptions of having one’s appearance commented on by professional counterparts, and unprofessional addresses were described as being addressed in ways that were either unprofessional or violated the norms within the courtroom.

### 5.1. Reported Sources of Negative Acts

Figure 1 shows the distribution of types of negative acts across judges and defense lawyers as reported sources. Overall, there appeared to be a tendency for prosecutors to report more illegitimate tasks, stressors, and verbal abuse from judges and more general- and person-focused incivility from defense lawyers.

### 5.2. Gender

Table 2 presents the distribution of reported negative acts across genders (men/women). Statistically significant differences between men and women were found for illegitimate tasks and verbal abuse, with women reporting more experiences with both illegitimate tasks and verbal abuse compared to men.

### 5.3. Seniority

Table 3 presents the distribution of reported negative acts across seniority (assistant prosecutor/prosecutor or higher), with the Chi^2^ test showing a statistically significant difference between assistant prosecutors and more senior prosecutors for negative acts in the form of lack of court management and verbal abuse. Assistant prosecutors more frequently report experiencing verbal abuse compared to their more senior colleagues, who conversely more frequently report experiencing illegitimate stressors due to a lack of court management.

## 6. Discussion

The purpose of the present study was twofold: to describe and characterize the types of negative acts reported by Danish prosecutors in a career perspective and to examine possible differences in the source and distribution of these acts across demographic characteristics. We found that the 687 descriptions of incidents with negative acts reported by Danish prosecutors could be characterized by 12 categories within three overarching themes. The negative acts identified included illegitimate tasks and stressors in addition to illegitimate behaviour and incivility. Women more often reported verbal abuse and illegitimate tasks from professional counterparts compared to men. Verbal abuse was more often reported by less experienced prosecutors compared to their more experienced colleagues, who reported a lack of court management more often. Illegitimate behaviour was the largest of the main coding categories representing 89.1% of the reported experienced negative acts while illegitimate stressors and -tasks represent 9.9% and 3.2% of the described negative acts in the present study respectively.

The rates of exposure to negative acts among Danish prosecutors found in the present study were comparable to the rates among American attorneys found by Cortina and colleagues [29]. We found 64% of the Danish prosecutors had experienced unpleasant encounters with professional counterparts in court during their career with an average seniority of 10 years. This is in line with Cortina and colleagues [29] who found that 62% of their surveyed attorneys had experienced interpersonal mistreatment during litigation in the last five years. General incivility such as disrespect or professional discrediting was the most frequent type of negative act in both the present study and the study by Cortina and colleagues [29]. The similarities in frequency and type of negative acts could be indicative of general work environmental characteristics within courtrooms rather than a country-specific issue. Negative acts as a general characteristic of the work environment in courtrooms was suggested by Omari and Paull [22] describing the legal profession as characterized by a heightened level of conflict and incivility. This is further supported by the following five types of experiences that were replicated from Cortina and colleagues [29] in the current study: disrespect/dishonesty, professional discredit, threats/intimidation, unprofessional address, and appearance comments. However, minor differences were found between the current study and Cortina and colleagues’ [29] results, as ignoring or exclusion and silencing is combined in the present study because descriptions of being interrupted, talked over, or ignored during litigation were largely overlapping in a Danish context (for example: ‘… he ignored me 7 times, after which he said “I don’t feel like answering your question”’). Therefore, we combined the two concepts as they mostly represented overlapping aspects of Danish prosecutors’ experiences with being actively ignored or silenced in court. Additionally, Cortina and colleagues [29] reported ‘gender-related incivility’ as a separate category. Comparably, a broader category of ‘person-focused incivility’ was found to more accurately represent the experiences reported by Danish prosecutors in the present study. We found examples of behaviour targeting the prosecutors on a personal level were not confined to gender but were more diverse and mostly related to age and appearance instead. Specifically, our data suggest that age disparagement is more frequent in a Danish context (89.2%) compared to gender disparagement (16.2%) and that gender and age disparagement sometimes co-occur (‘... a judge just doesn’t like younger women’).

In the present study, we introduced verbal abuse as a subcode within illegitimate behaviour, aiming to include descriptions of high arousal such as being verbally abused, scolded, and yelled at by professional counterparts. The construct of verbal abuse is applied in the present study as a supplement to incivility as our data shows descriptions of high arousal. Incivility is defined as ‘low intensity’ by Andersson and Pearson [28] and existing literature on negative acts distinguish incivility from workplace bullying, as they are seen as low and high intensity behaviour respectively [17,34]. As such, verbal abuse would not be considered part of the original incivility construct due to its nature as high intensity [17,28]. Conversely, Cortina and colleagues [29] included a range of severity up to extreme hostility in their construct of general incivility, suggesting that uncivil behaviour is not incompatible with high intensity. Hence, verbal abuse could be considered a high intensity type of incivility that was not however directly coded by Cortina and colleagues [29]. This could point to a construct of incivility including a variety of intensity rather than being limited to low-intensity behaviour.

### 6.1. Differences in Demographic Characteristics and Perceived Source

The present study found few statistically significant differences in the distribution of negative acts across gender and seniority. Women more often reported having experienced illegitimate tasks and verbal abuse compared to men, which is similar to Cortina and colleagues’ [29] findings. However, whilst the current study only found statistically significant differences for two types of negative acts (illegitimate tasks, verbal abuse), Cortina and colleagues [29] found an overall difference in the reported experienced incivility. The limited differences in the experience of negative acts across gender in a career perspective found in the present study might suggest potential cultural differences between Denmark and USA. It is possible that more women in USA experience negative acts from professional counterparts compared to Denmark or more Danish men perceive acts from professional counterparts in court as negative compared to American men. In the present study, both defence lawyers and judges were reported as perceived sources of negative acts, but the rate of reported frequency differed based on the type of negative act (see Figure 1). Indications in our data suggest that negative acts revolving around work environmental stressors such as a lack of court management, a lack of resources, or having to perform work tasks outside of the prosecutors’ working area more frequently stemmed from judges rather than defence lawyers. Indeed, acts of this nature would be expected to stem most frequently from the judges, as they are the ones undertaking the court management. Furthermore, Cortina and colleagues [29] suggested that the occurrence of some incivility in court is due to ‘hard ball tactics’ used by defence lawyers, in which illegitimate behaviour is used as a strategy during proceedings. Several prosecutors in the present study described how they perceived the negative acts they experienced from defence lawyers to be with the purpose of rattling them during procedures, which support the suggestions of ‘hard ball tactics’ by Cortina and colleagues [29] in Danish courts of law. Furthermore, previous studies have suggested that deflecting the tricks and charades of the defence lawyers is a part of the prosecutor’s job [35]. 

Looking at negative acts across seniority, the analyses performed in the present study found several descriptions of negative acts referenced as stemming from the first years of the prosecutors’ career and rarely happening later. This points to more frequent reports of negative acts among the younger and less experienced assistant prosecutors, which is supported by the statistical analysis showing assistant prosecutors more frequently reporting experiencing verbal abuse from professional counterparts compared to their more experienced colleagues (see Table 3). Conversely, more experienced prosecutors more frequently reported experiences with a lack of court management. This could be a result of experienced prosecutors having participated in more court appearances compared to their junior colleagues, giving them a larger foundation of experience with courtroom procedures and what to expect when attending court. Such experience could lead to expectations concerning who oversees the court procedure and therefore the knowledge and experience to identify when the responsibility and managing of the court is lacking. As such, reporting and experiencing a lack of court management might only be available to more experienced prosecutors, who are able to identify when court management is missing, in opposition to less experienced prosecutors possibly mistaking a lack of court management for the customary court procedures.

### 6.2. Possible Antecedents of Negative Acts in the Danish Courtrooms

The present study considers only the prosecutors’ perspectives leaving out both the judges’ and the defence lawyers’ experiences of the interactions in court. In general, knowledge about perpetrators of negative acts is limited [36]. Several antecedents of negative acts perpetration have been suggested including work overload, imbalance in demands and resources, and an inactive and laissez faire management style [36]. In the context of the present study, judges were more frequently reported as a source of illegitimate tasks and stressors. Given the relatively low amount of general governmental expenditures spend on courts of law in Denmark [3] judges could be subjected to work overload as they, like the prosecutors, are employed by the Danish Government. If judges in Danish courtrooms are subjected to increased workload, demands, and a lack of resources, this may compromise their ability to manage interactions between professionals in court. A report on the working conditions for Danish judges made in 2020 by the Danish Association of Judges points to court management as a core task being hindered by competing work tasks that were previously taken care of by other members of staff [37]. This points to a perceived lack of court management caused by an increase in workload and a lack of resources. Such a lack of management due to increased work tasks could be perceived as an inactive or laissez faire management style by both defence lawyers and prosecutors, leaving more room for negative acts without consequences [36]. Applying negative acts as a strategy during court proceedings might be another cause of negative acts from defence lawyers, as was suggested by Cortina and colleagues [29]. A third cause of negative acts could be perceived assaults to the professional integrity of the legal professionals attending court. In a study of career motivations of state prosecutors, Wright and Levine [35] interviewed 267 prosecutors in the American Southeast and Southwest between 2010 and 2013. From this, they identified 4 narratives on career motivations for working state prosecutors. One of these narratives concerns the motivation to reinforce one’s core absolutist identity, and this was invoked by 90 of the interviewed prosecutors [35]. This narrative on core absolutist identity emphasizes discouraging questioning of the rules as parts of the prosecutor’s job and express the prosecutors’ intrinsic commitment to rules, structure, and categories of right and wrong. Furthermore, prosecutors are seen as valuing order and accountability, and as people who react to violations of rules with moral or righteous indignation. Such a narrative of a core absolutist identity is suggested by Wright and Levine [35] to also apply to other types of legal professionals such as defence lawyers. If so, defence lawyers might act with indignation and perceive negative acts in the court room as a way of defending the rules, structure, order, and accountability of the judicial system, which they otherwise perceive at risk of being compromised by prosecutors’ behaviour in court. Thus, if defence lawyers or judges perceive the way in which the prosecutors perform their court proceedings as a threat to the integrity of the legal system or profession, violating the order and structure of the court room, they might respond with incivility or verbal abuse towards the prosecutors. Nevertheless, future studies are needed to explore how both judges and defence lawyers perceive the social interactions between the professional counterparts in court.

### 6.3. Implications for Prevention

The characterization of negative acts experienced by Danish prosecutors created in the present study builds on the pilot study by Vang and colleagues [14], increasing our knowledge of the specific types of negative acts occurring in Danish courtrooms. By exploring the various types of negative acts reported by Danish prosecutors and identifying illegitimate behavior and general incivility as the most frequently reported types of negative acts, this study increases our knowledge of the working environment in the Danish legal system. Thus, the present study sheds light on a still underrepresented work group in the existing literature on negative acts—both in Denmark and internationally [22]. Knowledge of the type and distribution of negative acts can inform interventions on several levels.

Within the Danish Prosecution Service, findings from the current study can inform training initiatives for assistant prosecutors on what to expect during litigation as well as inform supervisors and mentors about what types of acts their trainees may need support in handling and processing. Especially less experienced prosecutors and women seem to more often report specific types of negative acts compared to their male or more experienced colleagues. Therefore, a focus within the Danish Prosecution Service on preparing the less experienced prosecutors or women for the tone of voice in court and supporting them in handling situations with perceived negative acts from their professional counterparts could prove useful in heightening their occupational mental health. Between organizations, findings from the current study can be used for educational purposes on what types of behavior are occurring and perceived as negative. This is potentially useful in establishing joint agreements between prosecutors, judges, and defense lawyers on acceptable and unacceptable behavior in court.

Another suggestion for preventing negative acts in courtrooms is to look into lowering the workload and heightening the resources within the Danish legal system. Increasing the judges’ ability to intervene during court proceedings and actively manage the court could possibly decrease the number of negative acts between defense lawyers and prosecutors. Here, ensuring the court management training provided to judges during their education includes sufficient focus on conflict and process management could prove helpful in heightening their ability to intervene in situations where the character of the conflicts changes from what would be expected during court proceedings to being characterized by negative acts. As such, interventions concerning the working environment in the Danish legal system could benefit all legal professionals, rather than being limited to the Danish prosecutors.

### 6.4. Methodological Considerations and Directions for Future Research

Significant overlap between the results from the present study and the results from Cortina and colleagues [29] points to comparisons in negative acts experienced in courtrooms across countries. However, the cultural similarities between the two countries should be considered, as they could explain the overlapping characteristics of negative acts identified in the two studies. Despite the similarities, differences in the workings of the legal systems in Denmark and America would be expected, but the present study and Cortina and colleagues’ [29] still present similar results. Thus, the negative acts experienced in American courtrooms resemble those experienced in Danish courtrooms. The experiences reported by prosecutors and analyzed in the present study were based on experiences from a career perspective. This resulted in a mean career length of 10 years with a standard deviation value of 9, and the findings from the current study are therefore reflective of the experiences of prosecutors throughout a period of 20 years. As such, the results from this study are not necessarily representative of the current state of Danish courts of law. The coding procedure involved coding the reported negative acts independently of when the experience occurred, whereby the current seniority and the seniority at the time of the negative act might not correspond. This should be considered when interpreting the statistical analysis of negative acts distributed across seniority. The sample size (N = 572) and response rate (64.0%) of the current study are high and are based on all Danish prosecutors being invited to participate. As such, the present study covers a large part of the population of interest, indicating a high degree of representativity and generalisability of the results. This strengthens the study and the presented results from both the statistical analyses and the directed content analysis, extending our knowledge of the negative acts experienced by Danish prosecutors. Despite the current study having a large sample size, the sample sizes within the subtypes of negative acts range from small samples of 22 reported incidents to large samples of 308 reported incidents. The small sample size present in most subtypes might add to the lack of statistically significant differences between demographic data; therefore, the current study comments on possible tendencies as well as statistically significant differences between groups. When asked to elaborate on experiences with negative acts, the prosecutors were given the following examples of negative acts: verbal abuse; a condescending attitude; offensive, derogatory, or devaluating personalized comments; disrespectful remarks or acts; or the like. This might have affected the types of experiences reported by the prosecutors. It is possible that the prosecutors have experienced more types of negative acts than the ones described and characterized in the present study. As such, the characteristics of negative acts presented here should not be seen as an exhaustive description of all negative acts occurring in Danish courtrooms, but rather as a thorough description of the ones reported. As the data collected for the current study is retrospective and relies on self-reported experiences with negative acts, there is a risk of recall bias possibly affecting the respondents’ answers. There is also a risk of selection bias, as some prosecutors (n = 350) have decided to elaborate on their experiences with negative acts while others (n = 16) have not. Therefore, a group of prosecutors have experienced negative acts but decided against elaborating on this in the survey. This could affect the results of the present study by either under- or overrepresenting the reported types of negative acts. The characteristics of negative acts presented here should by cautioned by this risk of bias. The present study expands the knowledge of negative acts characterized by the SOS theory to include legal professions such as prosecutors, thus adding insights from courtrooms to the existing knowledge on how negative acts affect the working environment. Therefore, the present study can inform future research. Specifically, prospective and longitudinal studies targeting the occurrence and frequency of negative acts are needed to build knowledge about the importance of negative acts compared to known organizational factors. This would also further inform and target preventive strategies and decrease the risk of under- or overrepresenting the reported types of negative acts. More longitudinal investigations into possible risk profiles for experiencing negative acts would also help develop intervention strategies.

## 7. Conclusions

The aim of the present study was to describe and characterize the negative acts from professional counterparts experienced by Danish prosecutors and to determine the source and distribution of the identified types of negative acts across demographic characteristics. Based on the SOS theory by Semmer and colleagues [23] and the construct of incivility introduced by Cortina and colleagues [29], we categorized negative acts into a total of 12 categories organized in three overarching themes of illegitimate tasks, stressors, and behavior. Illegitimate behavior and general incivility were the most common types of negative acts reported by Danish prosecutors. Differences in the distribution of negative acts across demographic data found women reporting experiencing verbal abuse and illegitimate tasks more often compared to men. Assistant prosecutors reported verbal abuse more often compared to higher-ranking colleagues, who conversely reported a lack of court management more often. As these differences were based on self-reported and retrospective data, they are not necessarily indicative of the current status of the working environment in Danish courts of law. Future research is required to identify the occurrence, distribution, and consequences of exposure to different types of negative acts from professional counterparts in court.

## Figures and Tables

**Figure 1 behavsci-14-00332-f001:**
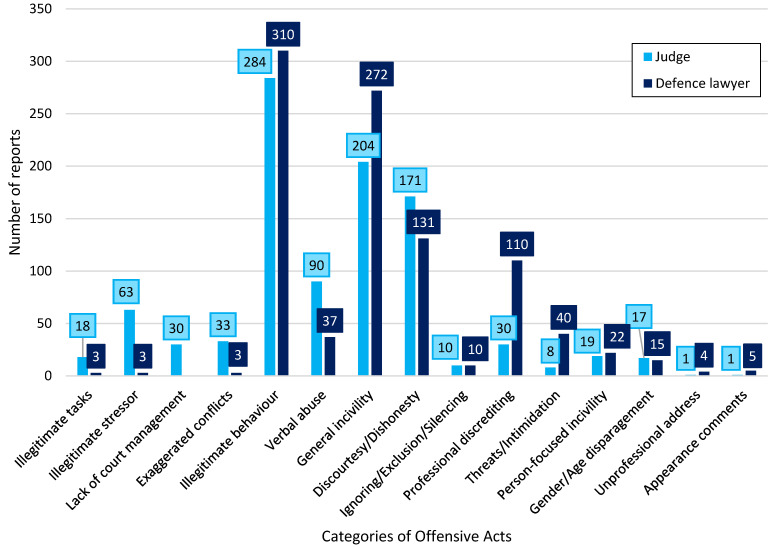
Bar chart showing the distribution of reported negative acts across judges (n = 312) and defense lawyers (n = 321) as the reported source. Note: Due to the responsibility for managing the court during proceedings, judges are presented as the only source of lack of court management.

**Table 1 behavsci-14-00332-t001:** Identified main- and subcodes for negative acts within the SOS framework.

Main- and Subcodes	%	Examples
**Illegitimate task**	3.2%	‘When you have to play-pretend as a janitor and move the chairs and see to the IT in court’. ‘A judge asked me to go outside and ask the municipality to stop cutting the grass, because it was noisy’. ‘Judges remarking on their appendix lacking numbering, is not clipped together, etc.’
**Illegitimate stressor**		
Lack of court management	4.4%	‘At one point, a defense lawyer almost started to interrogate me without the judge intervening, even though I called it to the attention of the court’. ‘A lack of court management resulting in the defense lawyer being allowed to verbally abuse the prosecutor …’.
Conflicts arising or exaggerated by illegitimate stressors	5.5%	‘A judge who yells or scolds—even though it isn’t the prosecutor’s fault that the problem has arisen’. ‘Verbal abuse from a judge due to a missing mental examination that another prosecutor had decided wouldn’t be necessary’.
**Illegitimate behavior**		
Verbal abuse	18.0%	‘A judge who … thrashed the pile of appendices to the floor, yelled “we won’t begin until this has been cleaned up” and left the room’. ‘I have experienced being verbally harassed by a defense lawyer, who pounded in the table with his hand and screamed that he bloody didn’t want to be interrupted’. ‘A judge yelled at me and called me offensive things in front of the defense lawyers’.
General incivility		
*Disrespect or dishonesty*	44.8%	‘Judges and defense lawyers who roll their eyes at me’. ‘A defense lawyer outright lied in their procedure about what I had just said, it was extremely unpleasant’.
*Ignoring, exclusion, or silencing*	3.1%	‘An experienced defense lawyer interrupts and interferes in the opening hearing of the accused. The interference is irrelevant and unnecessary, the tone of voice is unpleasant and blaming, and it happens with the sole purpose of rattling me and destroying my plan for questioning’. ‘A judge refused to answer my question, as it didn’t suit him. He ignored me seven times, after which he at last said: “I don’t feel like answering your question”’.
*Professional discrediting*	20.7%	‘Defense lawyers insinuating that you have destroyed evidence or are not objective’. ‘I’ve had a judge ask me if we didn’t learn anything in law school anymore’.
*Threats or intimidation*	6.8%	‘A defense lawyer threatened to report me and an investigator to the Independent Police Complaints Authority’. ‘Defense lawyers who degrade the prosecutor and intimidate’.
Person-focused incivility		
*Gender or age disparagement*	5.4%	‘Both judges and defense lawyers have several times given personally and degrading comments, among these, especially comments concerning the fact that I’m a woman’. ’I have been called ’the young prosecutor’ several times’.
*Unprofessional address*	1.0%	’When the phrase ”you have” rather than “the Prosecution Service has” is used’ ’Judge asking if it is Huey, Dewey, or Louie that represents the Prosecution Service today (it is degrading)’.
*Appearance comments*	1.2%	‘As a young prosecutor, the defense lawyer said during a procedure, that my procedure was far off and that nothing else were to be expected when the prosecutor was a blonde’. ‘A defense lawyer: as you stand there in your dress, you look like someone from a classical painting’.

Note: **bold** denotes the main coding themes, followed by one or two subcategories. Underline denotes first subcategory, while *italics* denotes second subcategory.

**Table 2 behavsci-14-00332-t002:** Distribution of reported negative acts from professional counterparts across genders.

Negative Acts, Present	Men		Adj. Std. R	Women		Adj. Std. R	χ^2^, *p*
	n	%		n	%		
**Illegitimate tasks**	**<5**	**NA**	**−2.4**	**21**	**4.2**	**2.4**	**5.737, *p* = 0.017**
Illegitimate stressor							
Lack of court management	9	4.9	0.5	20	4.0	−0.5	0.278, *p* = 0.598
Exaggerated conflicts	11	6.0	0.4	26	5.2	−0.4	0.172, *p* = 0.678
Illegitimate behavior							
**Verbal abuse**	**17**	**9.3**	**−3.6**	**107**	**21.4**	**3.6**	**13.223, *p* < 0.001**
General incivility							
Discourtesy/dishonesty	90	49.2	1.4	215	43.0	−1.4	2.070, *p* = 0.150
Ignoring/exclusion/silencing	<5	NA	−0.8	17	3.4	0.8	0.663, *p* = 0.416
Professional discrediting	37	20.2	−0.2	105	21.0	0.2	0.050, *p* = 0.824
Threats/intimidation	10	5.5	−0.9	37	7.4	0.9	0.783, *p* = 0.376
Person-focused incivility							
Gender/age disparagement	12	6.6	0.8	25	5.0	−0.8	0.634, *p* = 0.426
Unprofessional address *	0	0.0	−1.6	7	1.4	1.6	2.589, *p* = 0.199
Appearance comments *	<5	NA	−0.9	7	1.4	0.9	0.843, *p* = 0.689

Note: Adj. Std. R: adjusted standardized residuals. **Bold** = values significant at *p* < 0.05. * = *p*-value from Fisher’s exact test due to an expected cell count less than 5. % refers to column percentage. n < 5 and the corresponding NA for percentage refer to censured counts to maintain anonymity.

**Table 3 behavsci-14-00332-t003:** Distribution of reported negative acts from professional counterparts across seniority.

Negative Acts, Present	Assistant Prosecutor		Adj. Std. R	Prosecutor or Higher		Adj. Std. R	χ^2^, *p*
	n	%		n	%		
Illegitimate tasks	8	4.3	1.0	14	2.8	−1.0	0.926, *p* = 0.336
Illegitimate stressor							
**Lack of court management**	**<5**	**NA**	**−2.2**	**27**	**5.4**	**2.2**	**4.759, *p* = 0.029**
Exaggerated conflicts	13	6.9	1.0	25	5.0	−1.0	0.948, *p* = 0.330
Illegitimate behavior							
**Verbal abuse**	**44**	**23.4**	**2.2**	**80**	**16.0**	**−2.2**	**5.017, *p* = 0.025**
General incivility							
Discourtesy/dishonesty	92	48.9	1.3	216	43.3	−1.3	1.762, *p* = 0.184
Ignoring/exclusion/silencing	6	3.2	0.1	15	3.0	−0.1	0.016, *p* = 0.900
Professional discrediting	35	18.6	−0.8	107	21.4	0.8	0.665, *p* = 0.415
Threats/intimidation	10	5.3	−1.0	37	7.4	1.0	0.941, *p* = 0.332
Person-focused incivility							
Gender/age disparagement	8	4.3	−0.8	29	5.8	0.8	0.649, *p* = 0.420
Unprofessional address *	<5	NA	0.9	<5	NA	−0.9	0.854, *p* = 0.399
Appearance comments *	<5	NA	−0.9	7	1.4	0.9	0.900, *p* = 0.690

Note: Adj. Std. R: adjusted standardized residuals. **Bold** = values significant at *p* < 0.05. * = *p*-value from Fisher’s exact test due to an expected cell count less than 5. % refers to column percentage. n < 5 and the corresponding NA for percentage refer to censured counts to maintain anonymity.

## Data Availability

Anonymized data can be made available upon reasonable request to the corresponding author.

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
