# Peer review of "Negative Acts in the Courtroom: Characteristics, Distribution, and Frequency among a National Cohort of Danish Prosecutors"

_behavsci, 2024, doi:10.3390/bs14040332_

Round 1

Reviewer 1 Report

Comments and Suggestions for Authors

This is an interesting paper, and novel in the way you have used the two theoretical constructs of SOS Theory and Incivility. In reading the paper, several times I thought of how those would work across other professional domains.

I have added comments on several pages of the draft - these relate to small gaps in information or clarity.

Comments on the Quality of English Language

I believe the article could be significantly improved by some diligent word-smithing. I found the English language a little clumsy in many places. In only a small number of cases was it the use of words I might not have chosen, rather my comments relate to sentence structure.

Reviewer 2 Report

Comments and Suggestions for Authors

Page 1

Line 21 - no apostrophe needed after colleagues (not possessive)

Line 25 - from a career perspective? - not sure what ''in a career perspective'' means

Line 33 - suggest full stop after legal system and delete ''as''

Line 35 - add 'the' before conviction

Line 37 - full stop after 'individuals'- delete ''and

Lines 38-39 - delete ''according to the section in the....'' and replace with ''under the Danish Administration of Justice Act (1922) (the Administration of Justice Act).

Lines 40-41 - rewrite - 'According to the EU Justice Score Board (Euro-41 pean Commission, Directorate-General for Justice and Consumers, 2022), Denmark took 1st place [among who else]? in number of incoming court-cases 40 per 100 inhabitants in the years 2012-2020.'

Page 2

Line 46 replace 'hosting' with 'promoting' or 'fostering''

Line 47 - ''government-funded judges and prosecutors''

Line 48 - replaced 'fully processed and heard'' with 'heard'' or 'determined''

Line 51 - replace 'such as' with ''íncluding' - because you have already used 'such as' in this sentence.

Line 55 - add hyphen or space between 'post' and 'traumatic''

Line 59 - delete ''In''

Line 60 - 'reported''

Line 65 - correlated and correlations

Liné 66 - ''suggests'?

Line 80 - suggests?'

Page 3

Line 114 - 'do not examine

Line 118 - 'its'' impact not 'their impact

Line 120 - were you only looking at physically in the court room and not outside it - to the profession more broadly

Line 138 - whereby? - can you clarify the meaning of the rest of the sentence after wherefore?

Lines 139-140 - 'function and should be performed by another type of worker.''

Line 146 - are you able to give an example of each type of illegitimacy? For example, and illegitimate task might be asking a prosecutor to make coffee for the boss or to clean the workplace kitchen?

Line 147 - consists

Line 161 - courtroom management style/approach?

Line 162 - parties present?

Line 164 - ''as engaging in differential treatment between defence counsel and prosecutors

Line 166 - delete - before stressors

Line 176 - Cortina et al?

Line 179 - Cortina et al?

Line 181 -  ''Using an iterative coding process, the authors identified 11 types of adverse behaviours from the replies of 483 attorneys to open-ended questions. These questions focused on the attorneys' personal experiences of interpersonal mistreatment within the past 5 years of their career which made the greatest impression upon them.''

Line 186 - Cortina et al?

Line 187 - general incivility  (or disrespect/dishonesty ,...... threats/intimidation), was the most frequent form of incivility. This was followed by gender....including unprofessional forms of address....

Page 5

Line 205 - types of negative acts

Line 218 - ''in the period 10 August 2021 to 12 October 2021''. New sentence ''All prosecutors employed...

Line 223 - Conference for Prosecutors

Line 223 - on 23 September 2021

Page 12

Line 447 - try and break this paragraph up into several paragraphs to improve readability

Page 13

Line 497 - stressors

Line 498 - 'Given the relatively low government expenditure on courts of law in Denmark....

Line 500 - you are linking the low government spend to judge work overload because they are employed by the Danish government - is this a valid link?

Line 502 - may compromise

Line 503 - A 2020 report by...on judges' working conditions suggests the core judicial task of court management is hindered by....

Line 530 - prosecutors' court proceedings - do you mean behaviour?

Line 559 - wouldn't the judge in the courtroom already have the ultimate power to intervene?

Line 560 - more actively manage the court

Line 562 - Ensuring judge training in court, conflict, and process management may decrease the number of negative acts....

Page 15 - from line 568 - consider breaking this paragraph into smaller chunks to improve readability

Page 16 

Line 625 - Based on a combination of SOS-theory (who and date) and the construct of incivility developed by Cortina et al (2002) ..

Line 627 - remove hyphens from before stressors and behaviour

Comments on the Quality of English Language

I have included my comments/suggestions in the response to the authors
